# Elicit and Enhance: Advancing Multimodal Reasoning in Medical Scenarios

## Abstract

Effective clinical decision-making depends on iterative, multimodal reasoning across diverse sources of evidence. The recent emergence of multimodal reasoning models has significantly transformed the landscape of solving complex tasks. Although such models have achieved notable success in mathematics and science, their application to medical domains remains underexplored. In this work, we propose $MedE^2$, a two-stage post-training pipeline that elicits and then enhances multimodal reasoning for medical domains. In Stage-I, we fine-tune models using a limited number of text-only data samples containing precisely orchestrated reasoning demonstrations to elicit reasoning behaviors. In Stage-II, we further enhance the model's reasoning quality using rigorously curated multimodal medical cases, aligning model reasoning outputs with our proposed multimodal medical reasoning preference. Extensive experiments demonstrate the efficacy and reliability of $MedE^2$ in improving the reasoning performance of medical multimodal models. Notably, models trained with $MedE^2$ consistently outperform baselines across multiple medical multimodal benchmarks. Additional validation on larger models and under inference-time scaling further confirms the robustness and practical utility of our approach.

## 1 Introduction

Medicine is a multifaceted endeavor. It requires clinicians to process vast amounts of information, including patient medical histories, physical examination findings, and laboratory test results, to diagnose conditions, formulate prognoses, and determine appropriate treatment plans. In many clinical settings, an iterative reasoning process that evaluates multiple possibilities with progressively accumulated clinical information is considered fundamental to medical practice (Adler-Milstein et al., 2021; Singh et al., 2022; Croskerry & Clancy, 2022). Recent advancements in multimodal large language models (MLLMs), such as OpenAI-o-series (OpenAI, 2024) and Gemini-2.5-Pro (Google, 2025), have significantly advanced complex task performance. These models employ scaling inference time and emulate reflective cognitive processes, pushing capabilities to unprecedented levels. In light of these advancements, we explore the question: *How can we effectively extend the strategy of multimodal reasoning to medical domains*?

We begin by evaluating the capabilities of different models within the medical domain, as such assessments provide a natural starting point for probing a model's medical knowledge and reasoning abilities. The results shown in the left panel of Figure 1 indicate that current multimodal models (i.e., general-purpose (Bai et al., 2025b; Chen et al., 2024b) and those specifically designed for medical tasks (Chen et al., 2024a; Li et al., 2023) ) demonstrate strong performance on relatively simple visual question answering tasks (Liu et al., 2021; Lau et al., 2018). However, performance declines markedly on more complex tasks (Yue et al., 2024a;b), which require deeper comprehension and advanced reasoning. Since effective reasoning models should be grounded in robust foundation models (Ye et al., 2025), we adopt general-purpose MLLMs as the basis for developing specialized medical reasoning models.

Previous research (DeepSeek-AI, 2025; Ye et al., 2025; Li et al., 2025) primarily relied on carefully curated, challenging datasets from mathematical and scientific domains to incentivize the model's reasoning capacity. In contrast, although medical practice encompasses numerous scenarios requiring reasoning, the available data remain limited, especially those integrating clinical information. To

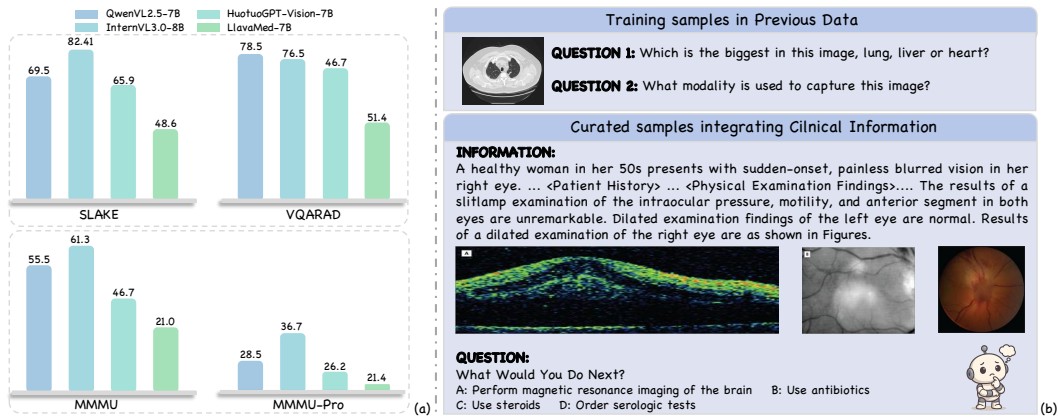

Figure 1: (a) Current models' performance on diverse tasks. (b) Samples that closely mirror real-world clinical scenarios are used to strengthen multimodal reasoning capabilities instead of samples focused primarily on pattern recognition or basic knowledge recall.

alleviate this, we adopt a rigorous data construction process involving meticulous data collection, cleaning, and expert review. Unlike earlier studies (Pan et al., 2025; Lai et al., 2025) that relied primarily on simple question–answer pairs solvable through memorized knowledge or pattern recognition (Figure 1), our curated dataset comprises 3K textual questions and 2K multimodal questions spanning 12 imaging modalities, including radiology, pathology, and optical coherence tomography. These samples are sourced from authoritative examinations (Jin et al., 2021) and publicly available case reports published in leading medical journals (PubMed, 2003). Each multimodal case integrates clinical histories, physical examinations, diagnostic investigations, and procedures.

With the dataset in place, we proceed to develop the models' reasoning capabilities specifically for clinical scenarios. Most of current work (Meng et al., 2025; DeepSeek-AI, 2025; Su et al., 2025; Li et al., 2025) usually employs rule-based reinforcement learning (Shao et al., 2024) to advance models' reasoning capabilities. However, challenges such as normalization of specialized medical terms (e.g., anatomical terminology) and hierarchical relationships (e.g., organ-system classifications) in medical domains make it difficult to verify responses solely through predefined rules. Moreover, the prevalent strategy that relies on outcome-based rewards is vulnerable to hallucinations during the reasoning process. This poses significant risks in clinical applications, where diagnostic and therapeutic decision-making demands exceptional logical rigor and strong evidence-based justification. Building upon these, we introduce a novel training pipeline, named $MedE^2$. Our pipeline involves two stages to progressively elicit and enhance the clinical cognitive reasoning capability of the models. Instead of *cold-start* on multimodal data (DeepSeek-AI, 2025), in the first stage, we conduct supervised fine-tuning on text-only data, where each sample includes carefully designed reasoning demonstrations to show how to utilize existing knowledge to solve complex tasks. Subsequently, we enhance the quality of the generated reasoning process to align with desired patterns. During this stage, we formulate Multimodal Medical Reasoning Preference (MMRP) and utilize Gemini-2.5-Pro to perform rejection sampling on our curated dataset, while employing Direct Preference Optimization (DPO) (Ziegler et al., 2019; Ouyang et al., 2022) to calibrate and refine the model's capabilities.

Extensive experiments are conducted to validate the effectiveness of our proposed pipeline. Despite being trained only on text-based reasoning data during Stage-I, the models demonstrate significant performance improvements across multiple medical benchmarks, achieving gains of at least 4.45% on MedXpertQA-MM (Zuo et al., 2025) and 6.67% on MMMU-Pro-Health (Yue et al., 2024b). These results highlight an innovative method for eliciting reasoning behaviors in multimodal medical tasks. Further enhancement in Stage-II enables models to exhibit superior reasoning capabilities, reaching performance competitive with several larger-scale models. We also show that improvements from $MedE^2$ generalize well to models with larger parameters. Moreover, the consistent improvements observed under inference-time scaling further demonstrate the robustness of our post-training recipe. In summary, our main contributions are as follows:

- We present $MedE^2$, a two-stage post-training pipeline to enhance multimodal reasoning in medical scenarios: (1) eliciting reasoning behavior by leveraging strategically developed text-only datasets, and (2) refining the reasoning quality by incorporating meticulously curated multimodal data.
- We construct a high-quality dataset of approximately 5,000 samples, comprising both text-based questions with reasoning chains and multimodal questions involving clinical information, thereby establishing a reliable corpus for bootstrapping medical reasoning.
- We perform comprehensive experiments across diverse benchmarks. Experimental results consistently demonstrate that $MedE^2$ significantly enhances model performance and achieves strong generalization across models of varying sizes, while also being robust to inference-time scaling.

## 2 RELATED WORK

### 2.1 MEDICAL MULTIMODAL LARGE LANGUAGE MODELS

The development of multimodal large language models has significantly advanced the field of medicine. Current medical multimodal large language models are primarily based on general multimodal models and are further trained on specialized medical datasets. This approach led to the emergence of numerous medical multimodal large models. For example, Med-PaLM (Tu et al., 2024) constructs the MultiMedBench dataset and fine-tunes based on PaLM-E (Driess et al., 2023). Recent studies continuously use similar strategies and refined training methods, resulting in significant progress, such as LLaVA-Med (Li et al., 2023), BioMedGPT (Zhang et al., 2024), MedTrinity-25M (Xie et al., 2024), and Med-Gemini (Saab et al., 2024). These strategies have demonstrated good application results in various medical scenarios, including medical dialogue (Ye et al., 2023), clinical decision support (Schubert et al., 2023), electronic health record analysis (Luo et al., 2022), and image report generation (Li et al., 2018). As an endeavor that involves multi-level analysis of details, the reasoning ability is vital for solving medical tasks. In this study, we aim to explore how to incentivize the reasoning abilities of multimodal large models in medical tasks.

### 2.2 REASONING MODELS

Enhancing the reasoning abilities of models remains a key challenge. Early efforts evolved from few-shot prompting to structured paradigms like Chain-of-Thought (Wei et al., 2022) and ReAct (Yao et al., 2023), aiming to emulate human-like reasoning. Subsequent work recognized reasoning as an iterative trial, error, and refinement process. The release of OpenAI's o1 (OpenAI, 2024) catalyzed further progress. Journey Learning (Qin et al., 2024) explored strategies for o1-style slow thinking and STILL-2 (Min et al., 2024) distilled long-form reasoning data to expand and refine solution paths. These advancements, culminating in DeepSeek's results (DeepSeek-AI, 2025), highlight the critical role of reinforcement learning in improving reasoning. Diverse reinforcement learning (RL) approaches to further bolster reasoning are now being actively explored and investigated (Wang et al., 2024; Wei et al., 2025). However, effective training strategies to enhance the reasoning abilities of medical models are still lacking. Previous research has primarily focused on constructing reasoning processes or adapting reinforcement learning (RL) frameworks (Su et al., 2025; Sun et al., 2025; Pan et al., 2025; Lai et al., 2025). In this paper, we propose a novel two-stage post-training recipe that progressively elevates models' ability to perform fine-grained reasoning in the medical domain.

## 3 PIPELINE

This section presents the core pipeline of $MedE^2$, as illustrated in Figure 2. To enhance the model's reasoning capabilities, we first curate a high-quality dataset comprising challenging text and multimodal question-answer pairs (Section 3.1). Based on our pilot studies, we select general MLLMs (e.g., QwenVL-2.5 (Bai et al., 2025b) and InternVL-3.0 (Chen et al., 2024b)) as the base models due to their performance across various medical tasks. Building upon these base models, we propose a two-stage training paradigm that progressively enhances the model's ability to reason in complex clinical scenarios (Section 3.2 and Section 3.3).

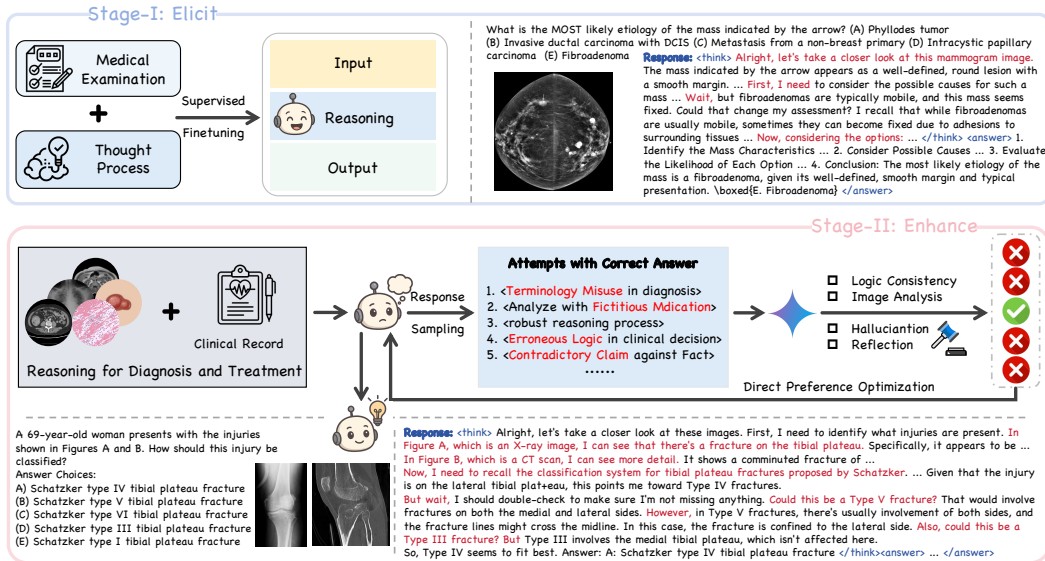

Figure 2: Overview of the two-stage post-training recipe *MedE*$^2$. In Stage-I, text-only data containing reasoning demonstrations is employed to elicit initial reasoning behavior. In Stage-II, Direct Preference Optimization is applied to multimodal data to further enhance reasoning quality.

## 3.1 DATA COLLECTION

We start with constructing a large-scale question bank sourced from authoritative examinations, top-tier journals, and publicly available datasets. In total, we collect 54K samples, consisting of 37K textual-only and 17K multimodal questions. To ensure high quality and prevent data leakage, a rigorous filtering process was implemented in collaboration with AI experts.

Given that our goal is to explore the potential of multimodal reasoning strategies in solving complex medical tasks, we prioritize filtering out relatively simple samples. Specifically, we employ two baseline models (i.e., QwenVL-2.5 and InternVL-3.0) to eliminate samples that could be solved directly. For multimodal questions, we further prompt Gemini-2.5-Pro to identify the question type and exclude those classified as pattern-recognition tasks. To ensure that the remaining questions are solvable, AI experts independently attempted each question, with 4 attempts per model (Gemini-2.5-pro (Google, 2025) with DeepSeek-R1 (DeepSeek-AI, 2025) for textual questions and Gemini-2.5-pro with Intern-S1 (Bai et al., 2025a) for multimodal questions). Only samples for which both experts answered correctly at least once were retained. Ultimately, this process resulted in a dataset consisting of 5K samples, including 3K textual and 2K multimodal questions.

## 3.2 STAGE-I: ELICITING REASONING ABILITY

To enable models to effectively reason in clinical problem-solving scenarios, it is essential to instruct them on utilizing their existing knowledge base to address complex reasoning tasks. Previous studies (Ye et al., 2025; Li et al., 2025) have demonstrated that while supervised fine-tuning with reasoning-specific data can enhance reasoning task performance, reasoning capabilities can be elicited effectively with only a few illustrative examples. Indeed, since much of the relevant knowledge has already been encoded during pre-training, the construction of precisely orchestrated demonstrations of reasoning processes is more critical than merely increasing the volume of training data. In this stage, we utilize textual-only reasoning demonstrations to elicit sophisticated reasoning capabilities, rather than relying on multimodal reasoning data. This choice is motivated by empirical findings (Su et al., 2025) that entangled multimodal reasoning data can impair the model's original language reasoning abilities. In contrast, training exclusively on textual data not only enhances reasoning skills but also maintains general visual understanding, albeit with a slight trade-off.

Building upon prior efforts (Qin et al., 2024; Huang et al., 2024; 2025), we adopt a distillation-based method to generate high-quality reasoning demonstrations using our curated dataset. Specifically, we

leverage state-of-the-art reasoning models, including Gemini-2.5-pro, DeepSeek R1, and Qwen3-235B (Team, 2025), to produce diverse solutions. For open-source models, we directly use their generated reasoning processes. For the proprietary model, such as Gemini-2.5-pro, whose intermediate reasoning steps are not readily accessible, we further prompt gpt-OSS-120B (OpenAI, 2025) to expand its summarized outputs into complete reasoning processes. We conduct rigorous evaluations that combine rule-based filtering with human-assisted validation to ensure the quality of generated solutions. We design three criteria: (i) the correctness of the final answer, (ii) logical structure and organization, and (iii) the plausibility and coherence of the reasoning process. Models trained on our curated textual reasoning data, denoted with the "`Stage-I`" suffix, demonstrate robust reasoning capabilities, not only in tackling textual domains but also when applied to multimodal scenarios.

### 3.3 STAGE-II: ENHANCING REASONING QUALITY

Although supervised fine-tuning can elicit reasoning behavior in models, it is often accompanied by an increase in hallucinations. This poses significant risks in medical scenarios, where clinical decision-making requires strong evidence-based justification. Prior work (Lv et al., 2024; Akbar et al., 2024) attributes this phenomenon to a mismatch between the generation paradigms during training and inference. Specifically, models are trained to predict the next token conditioned on preceding ground-truth tokens, while at inference time they rely on their own previously generated outputs. This discrepancy becomes particularly pronounced in tasks requiring long-form reasoning. As illustrated in Figure 2, even a correct answer can emerge from a flawed reasoning process involving terminology misuse, logical errors, or fabricated medical content, etc. To align model outputs with human preference, the mainstream method is to employ Reinforcement Learning from Human Feedback (RLHF) (Ziegler et al., 2019; Ouyang et al., 2022), exemplified by Preference optimization (PO). In this work, we introduce Multimodal Medical Reasoning Preference (MMRP) and integrate it with Direct Preference Optimization (DPO) (Rafailov et al., 2023; Xu et al., 2024). This combination effectively reduces hallucinations and generates reasoning processes that align more closely with user requirements.

**Preliminary** Considering a model as a policy $\pi_\theta(y|x)$ parameterized by $\theta$, RLHF aims at aligning the LLM $\pi_\theta$ with human preference. DPO is a representative algorithm of RLHF, serving as the preference loss to optimize the policy $\pi_\theta$ by enabling the model to learn the relative preference between chosen and rejected responses. DPO eliminates the requirement of training an explicit reward model based on the assumption of the Bradley-Terry model (Huang et al., 2004) and directly optimizes $\pi_\theta$:

$$\mathcal{L}_{\text{DPO}}\left(\pi_\theta\right) = -\mathbb{E}_{(\mathbf{x},\mathbf{y}_w,\mathbf{y}_l)\sim\mathcal{D}}\left[\log\sigma\left(\beta\log\frac{\pi_\theta\left(\mathbf{y}_w\mid\mathbf{x}\right)}{\pi_{\text{ref}}\left(\mathbf{y}_w\mid\mathbf{x}\right)} - \beta\log\frac{\pi_\theta\left(\mathbf{y}_l\mid\mathbf{x}\right)}{\pi_{\text{ref}}\left(\mathbf{y}_l\mid\mathbf{x}\right)}\right)\right]$$

where $\mathbf{y}$ denotes the response of $\mathbf{x}$, and $\mathbf{y}_w$, $\mathbf{y}_l$ represent the "win" and "lose" items in preference pairs. $\sigma$ is the sigmoid function. $\pi_{ref}$ is the reference model used to regularize $\pi_\theta$ via Kullback–Leibler divergence, with $\beta$ controlling the strength of regularization. DPO directly assigns higher probabilities to preferred responses, aligning with human preferences, while bypassing the train-inference mismatch present in SFT.

---
**Multimodal Medical Reasoning Preference**

- The reasoning process is logically coherent and step-by-step valid.
- Contain appropriate analysis of relevant visual information.
- Avoid introducing hallucinated content not grounded in the input.
- Includes self-checking, verification, or reflection on uncertainties.
---

**Preference Data Construction**. Initially, we prompt the Stage-I model $M_1$ to generate multiple candidate reasoning processes for each sample in our multimodal training dataset. Given an image $I$ and a question $q$, we sample candidate reasoning processes $y$ from the distribution $M_1(y \mid q, I)$, repeating this process 8 times per sample to form a candidate set $Y$. Notably, each question $q$ undergoes meticulous refinement through collaboration between human experts and a model (GPT-4o), explicitly removing descriptions of image $I$ from $q$. This ensures that reasoning arises from actual image examination rather than from textual captions. Subsequently, we filter out samples with wrong conclusions and employ Gemini-2.5-Pro to evaluate the remaining candidate reasoning

processes according to four MMRP criteria: Logical Consistency, Image Analysis Involvement, Absence of Hallucinations, and Presence of Reflection. To validate the correctness of the model judgment, we select 1,000 cases and recruit human annotators to assess them using the same criteria. The differences between human scores and model judge scores are presented in Figure 3.

We find that model and human evaluations are largely consistent. Reasoning processes meeting all criteria form the positive set ($Y_p$), whereas those failing any criteria constitute the negative set ($Y_n$). We then construct preference pairs by selecting a preferred response $y_c$ from $Y_p$ and contrasting it with a response $y_r$ from $Y_n$. Unlike static dataset construction, our MMRP methodology enables dynamic and flexible curation of multimodal preference datasets, effectively balancing specialization and generalization in reasoning tasks. The prompt used for data evaluation with Gemini-2.5-Pro can be found in Appendix A. Consequently, we obtain 4,432 pairs for QwenVL2.5-7B and 5,224 for InternVL3.0-8B. After applying DPO-based fine-tuning to the constructed preference dataset, the model demonstrates improved alignment with desired reasoning patterns and shows enhanced performance across diverse multimodal clinical scenarios.

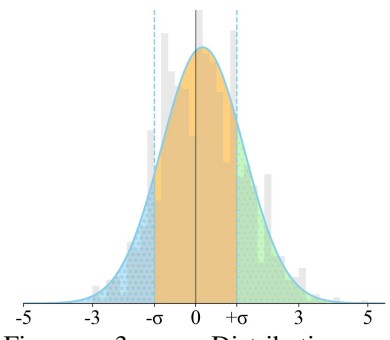

Figure 3: Distribution of human–model score differences, with 68.9% falling within $\pm\sigma$, where $\sigma=1.02$.

## 4 EXPERIMENTS

### 4.1 EXPERIMENTAL SETUP

We select QwenVL2.5-7B and InternVL3.0-8B as the base models to evaluate the effectiveness of $MedE^2$, resulting in two variants: (1) +Stage-I, which involves supervised fine-tuning on text-only data containing reasoning demonstrations; (2) +Stage-II, where the models from Stage-I are further enhanced using multimodal data and trained with DPO. We report the performance of GMAI-VL-R1 (Su et al., 2025) and Chiron-o1 (Sun et al., 2025) as baseline methods for comparison. We also benchmark our approach against state-of-the-art open-source models that operate at substantially larger parameter scales (Bai et al., 2025b; Chen et al., 2024b) and several leading proprietary models—GPT-4o, OpenAI-o1, Gemini-2.5-Pro, and QvQ-Max (Team). Details of Implementation can be found in Appendix C.

**Evaluation Benchmarks.** Our experiments involve two text-only tasks—MedQA (Jin et al., 2021) and Medbullets (Chen et al., 2025)—as well as three multimodal benchmarks: MedXpertQA-MM (Zuo et al., 2025), MMMU-Health (Yue et al., 2024a), and MMMU-Pro-Health (Yue et al., 2024b). Both MedQA and Medbullets are derived from questions used in the United States Medical Licensing Examination (USMLE). MedQA includes questions from both Step 1 and Step 2/3 of the Exam, while Medbullets focuses solely on Step 2/3, which are generally considered more challenging. For multimodal medical evaluation, MedXpertQA-MM serves as a highly challenging benchmark that assesses both medical knowledge and reasoning.

### 4.2 MAIN RESULTS

**Text-only SFT Elicits Reasoning Behavior.** As shown in Table 4.1, even a limited number of samples formatted with extended reasoning chains can effectively elicit the reasoning behavior, resulting in substantial performance gains. For example, QwenVL2.5-7B achieves improvements of +4.45% on MedXpertQA-MM and +6.67% on MMMU-Health. More impressively, the performance improvements from reasoning elicitation are even more pronounced for stronger base models. For example, InternVL3.0-8B achieves an 8% gain on MMMU-Health. This observation aligns with our hypothesis that training samples can serve as templates, illustrating how to utilize existing knowledge to solve complex reasoning tasks. Compared to GMAI-VL-R1 SFT, which relies on 10K multimodal samples, our method achieves a 6% increase on MMMU-Health and a 3.37% gain on MMMU-Pro-Health, using only half of the data in a text-only format. Similarly, relative to CHIRON-O1, which relies on large-scale supervised fine-tuning with constructed CoT data, $MedE^2$ exhibits a clear performance advantage across all benchmarks. This underscores that high-quality, task-targeted supervision is more critical than sheer data quantity for developing reasoning abilities.

Table 1: Results (%) of applying $MedE^2$ to QwenVL2.5-7B and InternVL3.0-8B on multimodal medical benchmarks, along with state-of-the-art methods and other training strategies. $\Delta$ indicates the performance gain over the base model, and the best improvements are highlighted in bold.

| Method | MedXpertQA-MM | | | | MMMU-Health | | MMMU-Pro-Health | |
|---|---|---|---|---|---|---|---|---|
| | Reasoning | Understanding | Overall | $\Delta$ | Overall | $\Delta$ | Overall | $\Delta$ |
| *Proprietary* | | | | | | | | |
| GPT-4o | 40.73 | 48.19 | 42.80 | – | 59.33 | – | 40.91 | – |
| QvQ-Max | 35.20 | 38.99 | 36.25 | – | 70.67 | – | 52.10 | – |
| OpenAI-o1 | 52.78 | 65.45 | 56.28 | – | 60.00 | – | 41.96 | – |
| Gemini-2.5-Pro | 61.69 | 69.13 | 63.75 | – | 80.67 | – | 66.08 | – |
| QwenVL2.5-32B | 26.00 | 30.86 | 27.35 | – | 63.33 | – | 48.25 | – |
| QwenVL2.5-72B | 27.10 | 31.58 | 28.35 | – | 70.00 | – | 50.35 | – |
| InternVL3.0-38B | 26.90 | 29.24 | 27.55 | – | 68.00 | – | 45.10 | – |
| InternVL3.0-78B | 28.80 | 35.92 | 30.20 | – | 71.33 | – | 51.40 | – |
| *Baselines* | | | | | | | | |
| QwenVL2.5-7B | 19.99 | 22.56 | 20.70 | – | 55.33 | – | 28.47 | – |
| InternVL3.0-8B | 21.09 | 23.83 | 21.85 | – | 61.33 | – | 36.71 | – |
| *Advancing* | | | | | | | | |
| GMAI-VL-R1 SFT | – | – | 23.55 | + 2.85 | 56.00 | + 0.67 | 32.99 | + 4.52 |
| GMAI-VL-R1 RLT | – | – | 23.80 | + 3.10 | 57.33 | + 2.00 | 34.03 | + 5.56 |
| +Stage-I (**ours**) | 24.48 | 26.90 | 25.15 | + 4.45 | 62.00 | + 6.67 | 36.36 | + 7.89 |
| +Stage-II (**ours**) | 25.80 | 28.52 | 26.55 | **+ 5.85** | 66.00 | **+ 10.67** | 38.81 | **+ 10.34** |
| Chiron-o1 | 23.30 | 25.10 | 24.20 | + 2.34 | 54.60 | - 6.72 | 33.90 | -2.80 |
| +Stage-I (**ours**) | 26.14 | 28.52 | 26.80 | + 4.95 | 69.33 | + 8.00 | 43.36 | + 6.65 |
| +Stage-II (**ours**) | 25.93 | 31.05 | 27.35 | **+ 5.50** | 70.00 | **+ 8.67** | 48.95 | **+ 12.24** |

**Multimodal DPO Enhances Reasoning Quality.** When Direct Preference Optimization (DPO) is applied after SFT in Stage-II, it builds upon the structured reasoning patterns established during the earlier stage and further enhances the model's output quality. Compared to Group Relative Policy Optimization (GRPO)-based tuning (Shao et al., 2024) (e.g., GMAI-VL-R1 RLT), our method achieves notably better results: 26.55% vs. 23.80% on MedXpertQA-MM, 66.00% vs. 57.33% on MMMU-Health, and 38.81% vs. 33.45% on MMMU-Pro-Health. A similar pattern is also observed in InternVL3.0-8B. These results suggest that merely forcing the model to reason over limited-solution-space tasks during training is suboptimal, often leading to hallucinations or incoherent reasoning. Instead, activating reasoning capabilities requires not only complex tasks but also training samples that engage the model in inference-time computation, or in other words, examples with precisely orchestrated solutions. Although the performance gain from DPO is slightly smaller than those from text-only SFT, we observe that DPO helps regulate the issue of "endless thinking" and encourages the model to "look before it thinks."

**Comparison with State-of-the Art Models.** Table 4.1 also presents a comparison with leading models. Among open-source models, those enhanced with $MedE^2$ demonstrate competitive or even superior performance despite having fewer parameters. For instance, on the MMMU-Health benchmark, QwenVL2.5-7B with $MedE^2$ achieves an accuracy of 66%, outperforming its larger counterpart, QwenVL2.5-32B. The performance of InternVL3.0-8B with $MedE^2$ has already surpassed InternVL3.0-38B and is slightly lower than InternVL3.0-78B. Similarly, on the MedXpertQA-MM and MMMU-Pro-Health benchmarks, the smaller models with $MedE^2$ exhibit performance comparable to that of larger-scale models.

## 4.3 ABLATION STUDIES

**Larger Models, Greater Improvements** Based on the preceding results, we observe that models exhibiting stronger initial performance tend to benefit more from reasoning elicitation. We hypothesize that larger models equipped with more extensive pretrained knowledge can derive increased benefits from our proposed $MedE^2$ framework. To rigorously verify this hypothesis, we apply our training pipeline to QwenVL2.5-32B and QwenVL2.5-72B and evaluate them on MedXpertQA-MM. We select MedXpertQA-MM for evaluation due to its increased challenge and minimal data leakage risk, as it was carefully constructed using difficulty-based filtering and data augmentation.

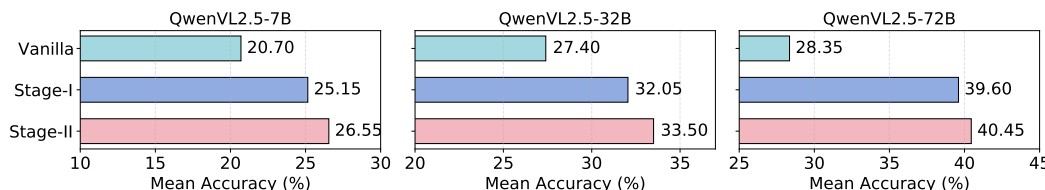

Figure 4: The performance of QwenVL2.5-7B, 32B and 72B on the MedXpertQA-MM benchmark. As the model size increases, the models demonstrate progressively greater benefits from the $MedE^2$.

Figure 4 illustrates the accuracy variations across different model sizes at various stages. Following Stage-I, QwenVL2.5-72B demonstrates the most substantial performance gain, outperforming both QwenVL2.5-32B (4.65%) and QwenVL2.5-7B (4.45%). These findings are consistent with our assumption in Section 3.2: models with more extensive prerequisite knowledge encoded within their parameter space are more capable of learning reasoning patterns from limited yet precisely structured exemplars. Combining the results shown in Table 4.1, we surprisingly find that QwenVL2.5-72B enhanced with $MedE^2$ even surpasses QvQ-Max (40.45% versus 36.25%) on the MedXpertQA-MM benchmark, which is currently one of the most powerful models within the Qwen series. Although larger proprietary models such as o1 and Gemini-2.5-Pro still maintain an advantage on challenging benchmarks, the application of $MedE^2$ significantly narrows the performance gap between open-source and proprietary models. These results further demonstrate the effectiveness of $MedE^2$ in advancing multimodal reasoning capabilities for medical tasks. Detailed experimental results can be found in Appendix D.

**Text-only vs. Multimodal Elicitation.** Previous studies have attempted to utilize multimodal training samples to activate multimodal reasoning capabilities. However, models optimized with such data often suffer from performance degradation on general vision tasks and impairment in linguistic abilities. Specifically, we compare text-only SFT, multimodal SFT, and their combination within Stage-I. In addition to evaluating on multimodal benchmarks (MedXpertQA-MM and MMMU-Health), we also assess performance on two language-focused benchmarks: MedQA and Medbullets. As illustrated in Figure 5, both text-only and multimodal SFT successfully elicit reasoning behaviors and improve overall performance. However, multimodal SFT yields relatively smaller improvements compared to text-only SFT. Notably, combining text-only and multimodal SFT results in reduced performance compared to the text-only SFT setting, particularly evident in the text tasks. This finding confirms that eliciting reasoning abilities through multimodal data may compromise the original language capabilities of the model.

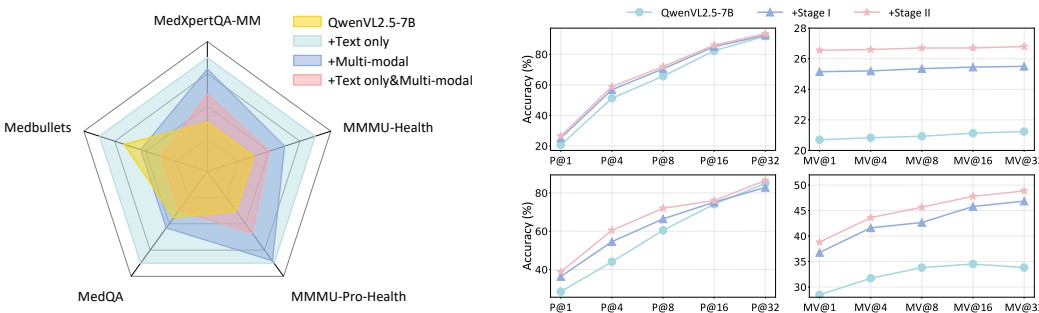

Figure 5: Comparison of performance on Med-XpertQA-MM using various strategies for eliciting reasoning behaviors.

Figure 6: The Accuracy of models with $MedE^2$ on MedXpertQA-MM (upper part) and MMMU-Pro-Health (lower part) with inference-time scaling.

**Benefit when Applying Inference Time Scaling.** To investigate the effectiveness of $MedE^2$, in scaling up inference time, we report the results of majority-vote accuracy and pass@n accuracy on MedXpertQA-MM and MMMU-Pro-Health across 32 computational runs with temperature=0.5. The results, illustrated in Figure 6, compare the performance of three models under Pass@n and majority voting (MV@n). Since both datasets are in a multiple-choice format with a limited answer

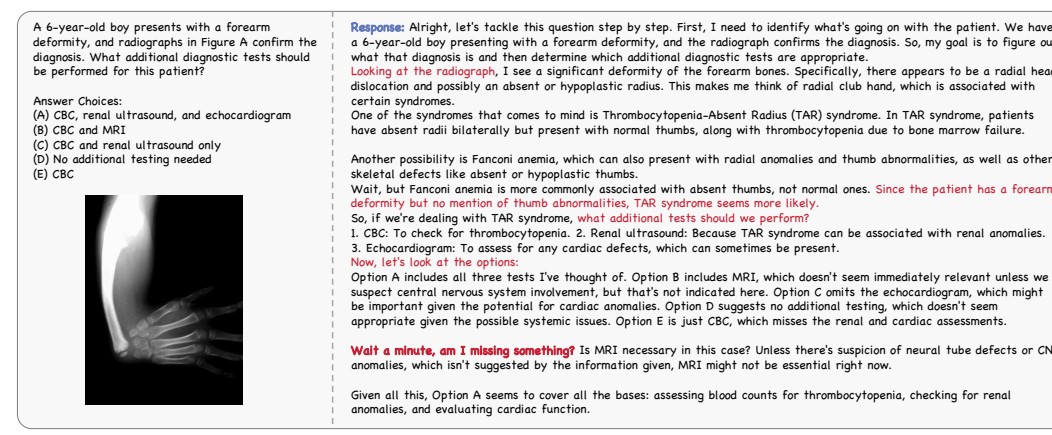

Figure 7: Case illustration of models' reasoning capability in tackling a clinical decision task.

space, model performance consistently improves as the number of attempts increases. Although the performance gain from majority voting on the MedXpertQA-MM dataset is modest due to its inherent difficulty, models trained with $MedE^2$ still demonstrate a significant and stable advantage as the number of computational attempts increases. On MMMU-Pro-Health, employing majority voting yields notable performance enhancements. Interestingly, the accuracy of QwenVL2.5-7B initially rises to a peak but then slightly declines as computational runs increase. In contrast, models trained with $MedE^2$ exhibit a steady performance improvement. These findings suggest that $MedE^2$ enhances model output consistency, particularly evident during majority voting.

### 4.4 QUALITATIVE ANALYSIS

Figure 7 demonstrates the reasoning capabilities of Our+Stage2 through its coherent analysis of determining appropriate follow-up examinations based on the patient's clinical presentation. In real-world scenarios, making an accurate diagnosis requires clinicians to integrate findings from multiple examinations to progressively narrow the differential diagnosis. Taking a closer look at the case, a notable observation is that the reasoning process is not constrained by the given options. Instead, it reflects a stepwise, hypothesis-driven diagnostic approach rather than merely discussing each option sequentially. Specifically, the model first identifies potential diagnoses (e.g., Thrombocytopenia-Absent Radius syndrome and Fanconi anemia) and then systematically excludes one by evaluating evidence from the radiographic findings. Decisions regarding subsequent examinations are logically derived from the provisional diagnosis, aligning closely with realistic clinical workflows. This observed capability highlights the potential of $MedE^2$ to serve as an effective post-training recipe to narrow the gap between current model performance and practical clinical application. Therefore, formulating an efficient and precise examination plan tailored to the patient's chief complaint is critical. We also remove the multiple-choice options and allow the model to respond freely, which can be seen in the Figure 8 in Appendix E.

## 5 CONCLUSION

In this study, we propose a novel reinforcement-learning-free training pipeline, $MedE^2$, designed to progressively elicit and enhance multimodal reasoning capabilities in medical domains. $MedE^2$ comprises two distinct stages: it first performs supervised fine-tuning on carefully curated text-only medical reasoning data, followed by preference-based optimization using multimodal data through Direct Preference Optimization. To support medical reasoning training, we constructed a dataset of 5K questions spanning both textual and multimodal clinical scenarios. Extensive experiments demonstrate the effectiveness and reliability of $MedE^2$ in advancing the reasoning abilities of medical multimodal models. The demonstrated scalability and consistent improvements under inference-time scaling highlight its broad applicability across different model architectures and parameter scales. We hope that this early exploration into multimodal reasoning for medicine will inspire further research into specialized clinical reasoning capabilities.

ETHICS STATEMENT

This work complies fully with the ICLR Code of Ethics. All datasets used in this paper are publicly available and were obtained from established open sources. No private, sensitive, or personally identifiable information was collected or used. The study involves no human subjects, no experiments on vulnerable populations, and no interventions requiring IRB approval. We confirm that our methodology and results do not raise foreseeable risks of harm, misuse, or ethical concerns beyond standard scientific research practices.

REPRODUCIBILITY STATEMENT

We will open-source the curated dataset and model weights at `https://anonymous.4open.science/r/MedEE-C381` to facilitate reproduction. We provide a comprehensive overview of our data construction methodology, the full processing pipeline, and the specific prompts used are provided in Appendix A. In addition, the training details and hyperparameter configurations for both stages of our method are presented in Appendix C.

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

# A  PROMPT TEMPLATES AND DATA FORMATS ADOPTED IN $MedE^2$

**The format of the text-only SFT data in Stage-I**

```
"messages": [
{
    "role": "system",
    "content": "You are a helpful assistant. A conversation between
    User and Assistant. The user asks a question, and the Assistant
    solves it. The assistant first thinks about the reasoning proc-
    ess in the mind and then provides the user with the answer. The
    reasoning process and answer are enclosed within <think> </think>
    and <answer> </answer> tags, respectively, i.e., <think>
    reasoning process here </think> <answer> answer here </answer>."
},
{
    "role": "user",
    "content": "{{Patient Information}} {{Question}}"
}
]
```

**Prompt used by the Preference Dataset creation process**

```
You are an expert reasoning evaluator.
I will provide you:
    * A question with images.
    * The groundtruth of the question
    * A model-generated answer, including its reasoning process.
Your task is to critically evaluate the reasoning based on
the following five aspects:
    1. **Answer Correctness**: Is the final answer correct based
    on the question?
    2. **Logical Consistency**: Is the reasoning process logically
    coherent and step-by-step valid?
    3. **Image Analysis Involvement**: Does the reasoning process
    demonstrate appropriate analysis of any visual information
    mentioned or implied in the question?
    4. **No Hallucination**: Does the reasoning process avoid
    introducing irrelevant or hallucinated information not supported
    by the question or known facts?
    5. **Reflection Presence**: Does the reasoning show
    any self-checking, verification, or reflection on
    possible uncertainties?
Please return the evaluation in the following JSON format:
    {
      "Answer_Correctness": "Yes/No",
      "Logical_Consistency": "Yes/No",
      "Image_Analysis_Involvement": "Yes/No",
      "No_Hallucination": "Yes/No",
      "Reflection_Presence": "Yes/No"
    }

<Question> \%s </Question>
<Groundtruth> \%s </Groundtruth>
<Answer> \%s </Answer>
```

Table 2: Performance comparison of QwenVL2.5-32B and QwenVL2.5-72B with *MedE*$^2$ across different task categories on the MedXpertQA-MM benchmark. As shown in the figure, the accuracy of each subtask improves, with the larger 72B model exhibiting a more pronounced increase compared to the 32B model.

| Method | Treatment | Basic Science | Diagnosis | Reasoning | Understanding | Overall | $\Delta$ |
|---|---|---|---|---|---|---|---|
| QwenVL2.5-32B | 28.13 | 25.78 | 27.61 | 26.07 | 30.87 | 27.40 | +0.00 |
| +Stage-I (**ours**) | 33.03 | 30.31 | 32.19 | 31.60 | 33.21 | 32.05 | +4.65 |
| +Stage-II (**ours**) | 35.93 | 32.57 | 32.86 | 33.54 | 33.39 | 33.50 | +6.10 |
| QwenVL2.5-72B | 30.36 | 24.08 | 28.86 | 27.11 | 31.59 | 28.35 | +0.00 |
| +Stage-I (**ours**) | 40.40 | 41.93 | 38.62 | 39.21 | 40.61 | 39.60 | +11.25 |
| +Stage-II (**ours**) | 41.52 | 42.49 | 39.45 | 39.63 | 42.60 | 40.45 | +12.10 |

## B LLM USAGE STATEMENT

We used LLMs to refine the writing, including checking grammar, rephrasing, and correcting typos. To ensure the writing quality, we further check and refine the generated text. In the experimental part of this study, LLMs were employed for two specific tasks: (1) Data preparation, as described in Section 3.2, and (2) Reasoning processes judgment, detailed in Section 3.3.

## C EXPERIMENTAL DETAILS

**Implementation Details.** All experiments are conducted on 8 NVIDIA H100 GPUs. LLaMA-Factory (Zheng et al., 2024) is used as the training framework, and vLLM (Kwon et al., 2023) serves as the backend during inference. For Stage-I, models are fine-tuned using LoRA (Hu et al., 2022) with a learning rate of $1 \times 10^{-4}$. Stage-II involves full-parameter fine-tuning with a learning rate of $1 \times 10^{-6}$, optimized using DeepSpeed (Rasley et al., 2020) with the ZeRO-3 configuration. Both stages adopt a cosine learning rate decay schedule. For models with larger parameters(i.e., 32B and 72B), we also employed LoRA in Stage-II.

## D EXPERIMENTAL RESULTS WITH DIFFERENT MODEL SCALES ON MEDXPERTQA-MM

Our method demonstrates more significant performance on larger-scale models, as shown in Table 2. These findings are consistent with our observations in Section 3.2: models with more extensive prerequisite knowledge encoded within their parameter space are more capable of learning reasoning patterns from limited yet precisely structured exemplars.

# E CASE STUDY

A 6-year-old boy presents with a forearm deformity, and radiographs in Figure A confirm the diagnosis. What additional diagnostic tests should be performed for this patient?

Correct Answer:
CBC, renal ultrasound, and echocardiogram

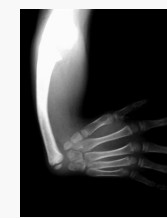

**Response:** Alright, let's think through this case carefully. I have a 6-year-old boy who presents with a forearm deformity. The radiograph confirms the diagnosis. I need to determine what additional diagnostic tests should be performed for him.

First, I need to identify what the diagnosis is based on the radiograph. The image shows a forearm with a single bone extending from the elbow to the wrist, which appears to be a radius-like structure. There is no ulna visible. This suggests that the boy has radial club hand due to congenital absence of the ulna (radial aplasia). Radial club hand is a congenital anomaly characterized by the absence or hypoplasia of the radius, leading to radial deviation of the wrist and hand. It's important to note that this condition can be associated with other anomalies, so a thorough evaluation is necessary.

Now, considering potential associations, I recall that absent radius can be part of several syndromes:
1. **Thrombocytopenia-absent radius (TAR) syndrome:** Characterized by absent radii and thrombocytopenia.
2. **Fanconi anemia:** Associated with absent radii and other skeletal anomalies, as well as increased risk of malignancies.
3. **VACTERL association:** Involves vertebral defects, anal atresia, cardiac defects, tracheoesophageal fistula, renal anomalies, and limb abnormalities.
4. **Holt-Oram syndrome:** Features heart defects and upper limb anomalies.
5. **Roberts syndrome:** Includes limb defects and facial anomalies.

Given these possibilities, I need to decide which additional tests would help identify any associated conditions.

**For TAR syndrome,** I should consider ordering a complete blood count (CBC) to check for thrombocytopenia.
**For Fanconi anemia,** a CBC would also be useful since it can show pancytopenia. Additionally, a bone marrow biopsy might be needed if there's suspicion of bone marrow failure.
**In VACTERL association,** imaging studies like echocardiography (for cardiac defects), abdominal ultrasound (for renal anomalies), and possibly MRI or CT scans (for vertebral and tracheoesophageal defects) would be appropriate.
**For Holt-Oram syndrome,** cardiac evaluation is crucial, so an echocardiogram would be indicated.
**Roberts syndrome** typically presents with facial anomalies and limb defects, but genetic testing would be more specific for diagnosis.

Wait, am I missing anything? Could there be other tests that are more immediately relevant? Considering that hematologic abnormalities are common in some of these syndromes, starting with a CBC seems prudent. It can detect thrombocytopenia (TAR syndrome) and pancytopenia (Fanconi anemia). If the CBC reveals abnormalities, further specialized tests can be ordered. Also, given the possibility of cardiac anomalies in Holt-Oram syndrome and VACTERL association, an echocardiogram is important. Similarly, renal ultrasound is essential to rule out renal anomalies seen in VACTERL and sometimes in TAR syndrome.

Therefore, the additional diagnostic tests that should be performed for this patient are:
**Order a complete blood count (CBC), Perform an echocardiogram, Conduct a renal ultrasound**

Figure 8: Case illustration of models' reasoning capability in the open-ended task.

A 69-year-old woman presents with the injuries shown in Figures A and B. How should this injury be classified?

Answer Choices:
(A) Schatzker type IV tibial plateau fracture
(B) Schatzker type V tibial plateau fracture
(C) Schatzker type VI tibial plateau fracture
(D) Schatzker type III tibial plateau fracture
(E) Schatzker type I tibial plateau fracture

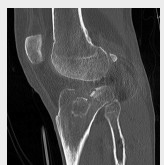

**Stage-I Response:**
Alright, let's take a closer look at these images. First, I need to identify what injuries are present.

In Figure A, which is an X-ray image, I can see that there's a fracture on the tibial plateau. Specifically, it appears to be a fracture of the lateral condyle of the tibia. There's a step-off deformity visible, indicating a significant displacement.

In Figure B, which is a CT scan, I can see more detail. It shows a comminuted fracture of the lateral tibial plateau with multiple fragments. The fracture lines extend into the articular surface, suggesting a high-energy injury.

Now, I need to recall the classification system for tibial plateau fractures proposed by Schatzker. He divided tibial plateau fractures into six types based on the location and pattern of the fracture:
   - **Type I:** Fractures involving the anterior tibial spine.
   - **Type II:** Fractures involving the posterior tibial spine.
   - **Type III:** Fractures involving the medial tibial plateau.
   - **Type IV:** Fractures involving the lateral tibial plateau.
   - **Type V:** Fractures involving both the medial and lateral tibial plateaus.
   - **Type VI:** Fractures involving the tibial tuberosity.

Given that the injury is on the lateral tibial plateau, this points me toward Type IV fractures.

But wait, I should double-check to make sure I'm not missing anything. Could this be a Type V fracture? That would involve fractures on both the medial and lateral sides. However, in Type V fractures, there's usually involvement of both sides, and the fracture lines might cross the midline. In this case, the fracture is confined to the lateral side. Also, could this be a Type III fracture? But Type III involves the medial tibial plateau, which isn't affected here.

So, Type IV seems to fit best. But I should also consider potential mistakes. For instance, if I misread the CT scan and thought there was involvement of the medial side as well, I might incorrectly classify it as a Type V fracture. Another common mistake is confusing the location of the fracture. If I mistakenly thought the injury was on the medial side, I'd choose Type III instead.

Therefore, after considering all this, I believe the correct classification is **A. Schatzker type IV tibial plateau fracture.**

**Stage-II Response:**
Alright, let's tackle this problem step by step. First, I need to carefully examine the images provided.

In Figure A, there's an X-ray of the knee joint. In Figure B, there's a CT scan showing the same area. Looking at Figure A, I notice that the tibial plateau appears to have a significant depression or collapse. There's a noticeable gap between the tibial plateau and the femoral condyles, indicating a loss of bone height. This suggests a fracture that has led to a collapse of the tibial plateau. Now, considering the CT scan in Figure B, it provides more detail. It shows a large, depressed fragment on the lateral side of the tibial plateau. The fragment is displaced downward and outward, which is characteristic of a lateral plateau fracture. So, putting this together, we have a lateral tibial plateau fracture with significant depression and displacement.

Now, I need to recall the classification system for tibial plateau fractures proposed by Schatzker. He divided tibial plateau fractures into six types based on the location and pattern of the fracture: ...

Given that our patient has a lateral tibial plateau fracture with significant depression and displacement, this fits best with **Schatzker Type III.**

Wait, but before I settle on that, I should consider whether any other options might fit better.

Option A is Schatzker Type IV, which involves the anterior and lateral portions. While there is some anterior involvement, the primary fracture line seems to be lateral, making Type IV less likely.

Option B is Schatzker Type V, which involves the posterior and lateral portions. Again, while there's some posterior involvement, the main fracture is lateral, so Type V doesn't seem to match as well.

Option C is Schatzker Type VI, which includes complex fractures involving multiple portions. Our patient has a single fracture line, so Type VI is unlikely.

Option E is Schatzker Type I, which involves the anterior and medial portions. There's no medial involvement here, so Type I doesn't fit.

Option D is Schatzker Type III, which matches our findings perfectly.

So, the correct answer is **D: Schatzker type III tibial plateau fracture.**

Figure 9: Case illustration of the differences in the responses of models in Stage-I and Stage-II. Blue text in the figure indicates visual hallucinations generated by the models.

