# Supplementary Material

Figure 1 shows the responses of QwenVL2.5-72B in Stage-I and Stage-II to a clinical diagnostic case from the MMMU-Health benchmark. Text highlighted in blue indicates visual hallucinations produced by the models. As shown, in Stage-I, the model incorrectly identified the image as indicating gallbladder cancer, which led to a cascade of erroneous reasoning. Notably, the model showed hesitation when analyzing the correct answer, as seen in the statement "Obesity is a risk factor for gallstones due to increased cholesterol saturation in bile." This issue was resolved in Stage-II.

A 48-year-old woman has experienced episodes of sharp abdominal pain for the past month. On physical examination there is tenderness upon palpation of her right upper quadrant. Laparoscopic surgery is performed and the gross appearance of her disease process is depicted here. Which of the following is the most likely risk factor for her disease? <image 1>

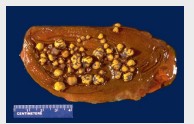

Answer Choices:
(A) Gilbert disease
(B) Cytomegalovirus infection
(C) Hemolytic anemia
(D) Ulcerative colitis
(E) Obesity

---

<think> Alright, let's tackle this question step by step. First, I need to analyze the image provided. It shows a section of what appears to be a gallbladder with multiple small, yellowish nodules on its surface. The gallbladder itself looks inflamed, with a thickened wall and a dark, possibly necrotic area at one end. Given that, I'm thinking this could be a gallbladder with gallstones and cholecystitis. The presence of gallstones is common, but the multiple small nodules are more characteristic of gallbladder cancer.

Now, considering the options:

**Option A: Gilbert disease** Gilbert disease is a benign condition characterized by increased unconjugated bilirubin levels due to decreased activity of UDP-glucuronosyltransferase-1 (UGT1). It doesn't typically cause gallbladder stones or cancer. So, this seems less likely.

**Option B: Cytomegalovirus infection** CMV infections can affect the liver and gallbladder, leading to inflammation and sometimes gallstones. However, CMV is more commonly associated with immunocompromised patients, like those with HIV/AIDS or organ transplant recipients. Our patient is a 48-year-old woman without mention of immunodeficiency. So, while possible, it's not the most likely risk factor.

**Option C: Hemolytic anemia** Hemolytic anemia can lead to increased red blood cell breakdown, which releases iron into the bloodstream. Excess iron can deposit in tissues, including the gallbladder, causing hemosiderosis. Hemosiderosis can increase the risk of gallstones and gallbladder cancer. This seems plausible.

**Option D: Ulcerative colitis** Ulcerative colitis is an inflammatory bowel disease affecting the colon. It doesn't have a direct link to gallbladder disease. While it can increase the risk of colorectal cancer, it doesn't explain the gallbladder findings.

**Option E: Obesity** Obesity is a risk factor for gallstones due to increased cholesterol saturation in bile. However, obesity isn't directly linked to gallbladder cancer. So, while obesity might contribute to gallstones, it's not the most significant risk factor for her disease.

Wait, am I missing something? Could obesity be indirectly related through other factors? Hmm, perhaps obesity increases the risk of gallbladder cancer by promoting chronic inflammation and bile stasis, which can lead to gallstones and eventually cancer. But still, obesity isn't listed as a direct risk factor among the options. Also, considering the multiple small nodules on the gallbladder wall, which are characteristic of gallbladder cancer, the most likely risk factor is hemolytic anemia leading to hemosiderosis.

Therefore, the best answer is **Option C: Hemolytic anemia.**
</think><answer>
1. **Analyze the Image**: The image shows a gallbladder with multiple small, yellowish nodules on its surface, indicating gallstones. The gallbladder also appears inflamed with a thickened wall and a dark, possibly necrotic area.
2. **Identify the Disease Process**: The presence of gallstones and gallbladder inflammation suggests cholecystitis. The multiple small nodules are characteristic of gallbladder cancer.
3. **Consider Risk Factors for Gallbladder Cancer**:
   – **Hemolytic Anemia**: Leads to increased red blood cell breakdown, releasing iron into the bloodstream.
   – **Iron Deposition**: Excess iron can deposit in tissues, including the gallbladder, causing hemosiderosis.
   – **Increased Risk of Gallstones and Gallbladder Cancer**: Hemosiderosis promotes the formation of gallstones and increases the risk of gallbladder cancer.
4. **Evaluate Other Options**:
   – **Gilbert Disease**: Benign condition with increased unconjugated bilirubin; does not cause gallbladder stones or cancer.
   – **Cytomegalovirus Infection**: More common in immunocompromised patients; not the primary risk factor.
   – **Ulcerative Colitis**: Inflammatory bowel disease affecting the colon; not linked to gallbladder disease.
   – **Obesity**: Increases risk of gallstones but not directly linked to gallbladder cancer.
5. **Conclusion**: Among the options, hemolytic anemia is the most likely risk factor for developing gallbladder cancer due to hemosiderosis
\boxed{C. Hemolytic anemia}
</answer>

---

**Stage-II Response:**

<think> Alright, let's tackle this problem step by step. First, I need to carefully read the question and examine the image provided. The patient is a 48-year-old woman who has been experiencing episodes of sharp abdominal pain for the past month. On physical examination, there's tenderness upon palpation of her right upper quadrant. They performed laparoscopic surgery, and the gross appearance of her disease process is shown in the image.

Looking at the image, I see what appears to be a gallbladder with multiple gallstones inside. The stones vary in size and color—some are yellowish, some are brownish. This suggests that she has cholelithiasis, or gallstones. Now, considering her symptoms: episodes of sharp abdominal pain and tenderness in the right upper quadrant are classic signs of biliary colic or acute cholecystitis due to gallstones obstructing the cystic duct. So, the question is asking for the most likely risk factor for her disease, which is gallstones.

Let me go through each of the options:

**Option A: Gilbert disease** Gilbert disease is a mild hereditary disorder characterized by increased levels of unconjugated bilirubin due to decreased activity of UDP-glucuronosyltransferase. Patients often have mild jaundice but usually don't form gallstones because the bilirubin remains unconjugated. So, Gilbert disease isn't a significant risk factor for gallstone formation.

**Option B: Cytomegalovirus (CMV) infection** CMV can cause various infections, especially in immunocompromised patients, but it's not associated with gallstone formation. Gallstones aren't typically linked to viral infections like CMV.

**Option C: Hemolytic anemia** Hemolytic anemia leads to increased breakdown of red blood cells, resulting in elevated levels of unconjugated bilirubin. This excess bilirubin can form pigment gallstones (black stones). However, the gallstones in the image appear to be cholesterol stones, which are more common and are yellowish in color. But wait, could some of these stones be pigment stones? Maybe I shouldn't rule out hemolytic anemia just yet.

**Option D: Ulcerative colitis** Ulcerative colitis is an inflammatory bowel disease affecting the colon and rectum. It's associated with primary sclerosing cholangitis, which can lead to bile duct strictures and gallstones. But primary sclerosing cholangitis is more commonly associated with Crohn's disease than ulcerative colitis. Also, there's no mention of liver function abnormalities or signs of cholangitis in the question.

**Option E: Obesity** Obesity is a well-known risk factor for gallstone formation, particularly cholesterol gallstones. In obese individuals, the liver produces excess cholesterol, which can precipitate in the gallbladder, leading to stone formation. The gallstones in the image are yellowish, consistent with cholesterol stones.

Wait, earlier I considered hemolytic anemia as a possible risk factor because of pigment gallstones. But the stones in the image seem predominantly cholesterol stones rather than pigment stones. Pigment stones are usually black or brown due to high bilirubin content. Also, the patient is 48 years old, which is within the typical age range for gallstone formation, especially in women. Women are more prone to gallstones due to hormonal influences, such as estrogen increasing cholesterol synthesis. So, putting it all together, obesity seems to be the most likely risk factor for her gallstones. Therefore, the correct answer is **E: Obesity.**
</think><answer>
1. **Identify the Disease Process**: The patient is a 48-year-old woman with episodes of sharp abdominal pain and tenderness in the right upper quadrant. The surgical specimen shows multiple gallstones in the gallbladder, indicating cholelithiasis (gallstones).
2. **Analyze the Gallstones**: The gallstones are yellowish in color, which is characteristic of cholesterol gallstones.
3. **Consider Risk Factors for Gallstones**:
   – **Obesity**: Increases cholesterol production in the liver, leading to supersaturation of bile with cholesterol and promoting gallstone formation.
   – **Hemolytic Anemia**: Leads to increased unconjugated bilirubin, forming pigment (black) gallstones.
   – **Gilbert Disease**: Causes mild unconjugated hyperbilirubinemia but does not significantly increase gallstone risk.
   – **Cytomegalovirus Infection**: Not associated with gallstone formation.
   – **Ulcerative Colitis**: Associated with primary sclerosing cholangitis but less commonly linked to gallstones compared to Crohn's disease.
4. **Match the Patient's Presentation with Risk Factors**: The patient's age, gender, and the type of gallstones (cholesterol stones) align with obesity as the most likely risk factor.
\boxed{E. Obesity}
</answer>

Figure 1: Illustration of the response differences of QwenVL2.5-72B between Stage I and Stage II from the MMMU-Health benchmark. Text highlighted in blue indicates visual hallucinations produced by the models.

Figure 2 displays QwenVL-2.5-72B's Stage-I and Stage-II answers to a basic-medical-knowledge item from the MMMU-Health benchmark. Examination of the blue text highlights three major misconceptions:

- **Location of pallidothalamic fibres.** The anterior limb of the internal capsule does not carry these fibres. Pallidothalamic axons arise from the internal segment of the globus pallidus (GPi) and either pierce the posterior limb as the lenticular fasciculus or sweep around it as the ansa lenticularis; their association is therefore with the posterior limb, not the anterior.
- **Subdivision of the internal capsule.** The capsule is classically partitioned into five regions—anterior limb, genu, posterior limb, retrolenticular part and sublenticular part—and the functional contents the model attributed to its anterior and posterior limbs are incorrect.
- **Laterality of pallidothalamic projections.** These axons remain ipsilateral: they do not cross between hemispheres. Instead, they traverse or skirt the posterior limb before merging into the thalamic fasciculus on their way to the VA/VL nuclei of the same-side thalamus.

All three issues are resolved in the Stage-II response.

Pallidothalamic axons cross the <image 1>

Answer Choices:
(A) Anterior limb of the internal capsule.
(B) Genu of the internal capsule.
(C) Posterior limb of the internal capsule.

---

**Stage-I Response:**

<think> Alright, let's tackle this question about the crossing of pallidothalamic axons. First, I need to recall where the internal capsule is located and what its different parts are. The internal capsule is a bundle of white matter fibers that connects the thalamus to the cerebral cortex. It's divided into three main parts:

 1. **Anterior limb:** This part connects the globus pallidus (GP) to the thalamus.
 2. **Genu:** This is the central part of the internal capsule.
 3. **Posterior limb:** This connects the putamen to the thalamus.

Now, the question is asking about the crossing of pallidothalamic axons. Pallidothalamic axons originate from the globus pallidus and project to the thalamus. So, I need to figure out which part of the internal capsule these axons cross.

Initially, I might think that since the globus pallidus is on one side and the thalamus is on the other, the axons would cross over at some point. But wait, the internal capsule is a bundle of fibers, not just a single pathway. So, I need to consider the structure of the internal capsule. Looking back, I remember that the anterior limb of the internal capsule is the part that connects the globus pallidus to the thalamus. Therefore, the pallidothalamic axons would cross over in the anterior limb. But hold on, I should double-check my understanding. Could there be any confusion with the genu or posterior limb? The genu is the central part of the internal capsule, and the posterior limb connects the putamen to the thalamus. Neither of these parts directly connects the globus pallidus to the thalamus. So, they wouldn't be the correct choice for the crossing point.

Therefore, the correct answer must be the anterior limb of the internal capsule. So, the correct answer is: **A: Anterior limb of the internal capsule.**
</think><answer>

 1. **Identify the Location of the Internal Capsule:** The internal capsule is a bundle of white matter fibers connecting the thalamus to the cerebral cortex. It is divided into three parts: the anterior limb, genu, and posterior limb.
2. **Understand the Function of Pallidothalamic Axons:** Pallidothalamic axons originate from the globus pallidus and project to the thalamus.
3. **Determine the Crossing Point:** The globus pallidus is on one side, and the thalamus is on the opposite side. Therefore, the axons must cross over somewhere between these two structures.
4. **Locate the Correct Part of the Internal Capsule:** The anterior limb of the internal capsule connects the globus pallidus to the thalamus. Thus, the pallidothalamic axons cross over in the anterior limb.
5. **Eliminate Incorrect Options:**
   – **Genu of the Internal Capsule:** This is the central part of the internal capsule and does not connect the globus pallidus to the thalamus.
   – **Posterior Limb of the Internal Capsule:** This connects the putamen to the thalamus, not the globus pallidus.
6. **Conclusion:** The pallidothalamic axons cross the anterior limb of the internal capsule.
\boxed{A. Anterior limb of the internal capsule.}
</answer>

---

**Stage-II Response:**

<think> Alright, let's tackle this question about the pallidothalamic axons and their crossing point. First, I need to recall the pathways involved in the basal ganglia circuitry. The pallidothalamic tract is part of the indirect pathway, which originates from the globus pallidus interna (GPi). So, the axons that we're focusing on are those projecting from the GPi to the thalamus.

Now, I should think about the anatomical course of these fibers. They leave the GPi and head toward the thalamus. But before they reach the thalamus, do they cross over to the opposite side? If so, where does this decussation occur? I remember that the pallidothalamic fibers do indeed decussate, forming the ansa lenticularis and the fasciculus lenticularis. These structures are part of the internal capsule. Specifically, the ansa lenticularis loops around the internal capsule, and the fibers eventually cross over. But where exactly do they cross? Is it at the anterior limb, the genu, or the posterior limb of the internal capsule? Let me consider each option:

    **Option A: Anterior limb of the internal capsule** The anterior limb is located between the caudate nucleus medially and the lentiform nucleus laterally. It contains fibers like the frontal pontine fibers and some corticospinal fibers. Does it contain the pallidothalamic fibers? Hmm, I don't think so. The pallidothalamic fibers don't pass through the anterior limb.

    **Option B: Genu of the internal capsule** The genu is the bend between the anterior limb and the posterior limb. It contains corticobulbar fibers heading to the cranial nerve nuclei. Again, not the pallidothalamic fibers.

    **Option C: Posterior limb of the internal capsule** The posterior limb is situated between the thalamus medially and the lentiform nucleus laterally. It contains several important fiber tracts, including the corticospinal tract, corticopontine fibers, spinothalamic tract, and crucially, the thalamic radiations.

Wait, but the thalamic radiations carry sensory information from the thalamus to the cortex, not motor fibers from the globus pallidus interna to the thalamus. Am I mixing things up? Hold on, perhaps I'm getting confused. The pallidothalamic fibers actually decussate within the posterior limb of the internal capsule. They form the ansa lenticularis and fasciculus lenticularis, which loop around the internal capsule and cross over in the posterior limb. So, despite my initial hesitation, the correct answer must be **Option C: Posterior limb of the internal capsule** But just to be sure, let me double-check for any potential mistakes. Is there any chance that the pallidothalamic fibers cross in the anterior limb or genu? Given their course, it doesn't seem likely. The anterior limb and genu are more associated with other fiber tracts. Therefore, I feel confident that the pallidothalamic axons cross the posterior limb of the internal capsule.
</think><answer>
To determine where the pallidothalamic axons cross, we need to understand their anatomical course:
1. **Origin:** The pallidothalamic fibers originate from the **globus pallidus interna (GPi)** as part of the indirect pathway in the basal ganglia circuitry.
2. **Course:** They exit the GPi and form the ansa lenticularis, which loops around the internal capsule. The fibers continue as the fasciculus lenticularis and cross over to the opposite side.
3. **Crossing Point:** The decussation occurs within the posterior limb of the internal capsule. This area is located between the thalamus medially and the lentiform nucleus laterally.
4. **Exclusion of Other Options:**
   – **Anterior limb of the internal capsule:** Contains fibers like frontal pontine fibers; not involved in the pallidothalamic tract.
   – **Genu of the internal capsule:** Contains corticobulbar fibers; also not involved.
5. **Conclusion:** Based on the anatomical path, the pallidothalamic axons cross in the posterior limb of the internal capsule。
\boxed{C. Posterior limb of the internal capsule.}
</answer>

Figure 2: Illustration of the response differences of QwenVL2.5-72B between Stage I and Stage II from the MMMU-Health benchmark. Text highlighted in blue indicates knowledge hallucinations produced by the models.

Section 4.4 of the main paper shows that InternVL3.0-8B + `Stage-II` reaches an accuracy of 48.95% on the MMMU-Pro-Health benchmark—outperforming GPT-4o, OpenAI-o1, and the similarly sized QwenVL2.5-7B + `Stage-II`, which scores 38.81%, a margin of more than ten percentage points. Figure 3 illustrates a benchmark case in which QwenVL2.5-7B + `Stage-II` answers incorrectly, whereas InternVL3.0-8B + `Stage-II` answers correctly.

Which phrase best describes the pathology? <image 1> <image 2>
Answer Choices:
(A) Acute demyelination  (B) Acute traumatic contusion
(C) Acute cerebral aneurysm (D) Acute encephalitis
(E) Acute hemorrhagic infarction (F) Acute subdural hematoma
(G) Acute embolic infarction (H) Subacute cerebral artery occlusion
(I) Chronic embolic infarction (J) Acute anterior cerebral artery occlusion

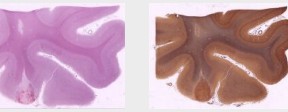

**QwenVL2.0-7B-stage-II:**

<think> Alright, let's take a closer look at these images. Image-1 shows a section of brain tissue stained with hematoxylin and eosin (H&E), giving it a pinkish hue. There are large areas that appear pale or empty-looking, possibly indicating regions where cells have been lost due to damage. In contrast, Image-2 is stained differently—maybe with something like myelin-specific stains—and shows darker brown areas interspersed with lighter regions. This suggests differences in tissue composition or staining affinity. First off, I need to figure out what these images represent. The H&E stain highlights cell nuclei and cytoplasmic components, while myelin stains target lipid-rich structures like myelin sheaths. So, Image-1 likely depicts neuronal loss, and Image-2 shows disrupted myelin.

Now, considering possible pathologies:

   **Option A: Acute demyelination** Acute demyelination would show loss of myelin without significant neuronal death initially. But wait, acute demyelination usually occurs over days to weeks, not immediately after injury. Also, it doesn't typically cause massive neuronal necrosis seen in Image-1.

   **Option B: Acute traumatic contusion** Contusions involve bruising of the brain tissue from trauma, leading to bleeding and swelling. That could explain some areas of pallor if there was ischemia, but wouldn't account for widespread myelin loss.

   **Option C: Acute cerebral aneurysm** An aneurysm rupture leads to subarachnoid hemorrhage, causing sudden bleeding into the brain. While this can cause infarcts, it wouldn't specifically highlight demyelination patterns.

   **Option D: Acute encephalitis** Inflammation from encephalitis affects neurons and glial cells, potentially causing necrosis. Again, not specific to demyelination.

   **Option E: Acute hemorrhagic infarction** This involves bleeding into an area of ischemia. Bleeding might obscure normal histology rather than showing clear demyelination.

   **Option F: Acute subdural hematoma** Similar to contusions, it results from blood accumulation under the dura mater, leading to pressure effects on the brain. Not directly related to demyelination.

   **Option G: Acute embolic infarction** An embolus blocks a vessel, causing ischemia and neuronal death. Neuronal loss fits, but does it affect myelin? Wait, maybe I'm missing something. Could an embolic event lead to demyelination? Emboli can cause both neuronal death and vascular changes affecting myelin.

   **Option H: Subacute cerebral artery occlusion** Subacute means over days to weeks. Our images seem more acute.

   **Option I: Chronic embolic infarction** Chronic processes don't match the acute appearance shown.

   **Option J: Acute anterior cerebral artery occlusion** This would affect the anterior part of the brain, possibly sparing other areas. Hold on, perhaps I should revisit

   **Option G: Acute embolic infarction.** An embolus can block small penetrating arteries supplying cortical layers, leading to both neuronal death and disruption of myelin tracts. This could explain the combination of neuronal loss and demyelination observed. But am I sure it's acute? Yes, because the lack of chronic changes supports an acute event. So, putting it all together, the most consistent explanation is acute embolic infarction, which accounts for both the neuronal necrosis and demyelination visible in the images.

Therefore, the best description is: **Answer: G: Acute embolic infarction**
</think><answer>

– **Image Analysis:**
   – Image-1 displays large areas of pallor indicative of neuronal loss.
   – Image-2 shows darkened regions corresponding to myelin-rich areas affected by pathology.
– **Pathological Interpretation:**
   – The presence of both neuronal death and demyelination suggests simultaneous damage to neural tissue and its supporting myelin sheaths.
   – Acute embolic infarction can cause ischemia leading to neuronal necrosis and disrupt myelin pathways.
– **Evaluation of Options: A: Acute demyelination:** Typically occurs over weeks; lacks significant neuronal loss. **B: Acute traumatic contusion:** Involves physical injury; less likely to cause demyelination. **C: Acute cerebral aneurysm:** Causes bleeding; does not specifically explain demyelination. **D: Acute encephalitis:** Involves inflammation affecting neurons and glial cells. **E: Acute hemorrhagic infarction:** Bleeding obscures histological details. **F: Acute subdural hematoma:** Results from trauma-induced bleeding. **G: Acute embolic infarction:** Can lead to both neuronal death and demyelination. **H: Subacute cerebral artery occlusion:** Occurs over days to weeks. **I: Chronic embolic infarction:** Fits with prolonged symptoms. **J: Acute anterior cerebral artery occlusion:** Specific regional involvement.
– **Conclusion:** Option G: Acute embolic infarction** best explains the combined findings of neuronal necrosis and demyelination in an acute setting.
**\boxed{G. Acute embolic infarction}**
</answer>

---

**InternVL3.0-8B-stage-II:**

<think> Alright, let's take a closer look at these images. Image-1 shows a section of brain tissue with a pinkish area and a darker central region, suggesting some active pathological process. Image-2 displays a brownish discoloration in a similar region, possibly indicating tissue breakdown or bleeding. Both images appear to be stained with hematoxylin and eosin (H&E), where pinkish hues often indicate edema or inflammation, and brownish or dark areas suggest hemorrhage or necrosis.

Now, time to interpret the significance. The pinkish area could represent acute tissue swelling due to inflammation or vascular leakage, while the darker or brownish regions point to tissue damage—possibly necrosis or hemorrhage.
Let's evaluate the diagnostic options one by one:

   - **Option A: Acute demyelination** – This usually affects white matter and presents with subtle changes; it doesn't produce the kind of gross hemorrhagic or necrotic appearance seen here.
   - **Option B: Acute traumatic contusion** – Trauma can cause focal bleeding (hemorrhage) and necrosis from mechanical injury. This fits the pinkish (edema) and brownish (necrosis or blood) pattern well.
   - **Option C: Acute cerebral aneurysm** – Ruptured aneurysms cause subarachnoid hemorrhage, not focal intraparenchymal damage as shown.
   - **Option E: Acute hemorrhagic infarction** – Ischemia with bleeding might cause this appearance, but it typically presents with irregular hemorrhage within infarcted regions. The pattern here feels more localized and consistent with trauma.
   - **Option F: Acute subdural hematoma** – This occurs outside the brain parenchyma, between meningeal layers, and would not cause direct tissue necrosis.
   - **Option G: Acute embolic infarction** – This would show ischemic necrosis, possibly pale or cavitated areas, but the presence of hemorrhagic components is less consistent unless it evolved into a hemorrhagic infarction.
   - **Option H: Subacute cerebral artery occlusion** – A subacute process would typically show early cavitation or chronic gliosis—not the acute changes seen here.
   - **Option I: Chronic embolic infarction** – Again, the features are too acute for this to fit.
   - **Option J: Acute anterior cerebral artery occlusion** – Infarcts in this region are possible, but the color and pattern still better match traumatic injury.
Putting this all together, the pinkish zone likely represents edema or early hemorrhage, and the brownish core suggests necrosis or clotted blood—hallmarks of brain contusion due to trauma.
Therefore, the best-fitting diagnosis is:
**\boxed{B. Acute traumatic contusion}**
</think><answer>

**Interpretation:**
- Pinkish zones reflect acute edema or inflammation.
- Brownish zones are likely necrosis or hemorrhage. These findings align with tissue damage from blunt force trauma.
**Option Evaluation:**
- **A: Acute demyelination** – Typically white matter; lacks necrosis/hemorrhage.
- **B: Acute traumatic contusion** – Trauma causes bleeding, edema, and necrosis. Fits both images.
- **C: Acute cerebral aneurysm** – Subarachnoid bleeding; not intraparenchymal like here.
- **D: Acute encephalitis** – Inflammation with scattered necrosis; less likely to look like this.
- **E: Acute hemorrhagic infarction** – Possible, but usually more irregular hemorrhage.
- **F: Acute subdural hematoma** – Extradural; not in brain tissue itself.
- **G: Acute embolic infarction** – Ischemia and possible necrosis, but less hemorrhage.
- **H: Subacute cerebral artery occlusion** – More chronic signs than seen here.
- **I: Chronic embolic infarction** – Inconsistent with acute findings.
- **J: Acute anterior cerebral artery occlusion** – May cause infarction, but doesn't fully explain the observed pathology.
**Conclusion:**
The most consistent diagnosis based on the pathology is:
\boxed{B. Acute traumatic contusion}
</answer>

Figure 3: Illustration of the response differences between QwenVL2.5-7B+`Stage-II` and InternVL3.0-8B+`Stage-II` from the MMMU-Pro-Health benchmark. Text highlighted in blue indicates knowledge hallucinations produced by the models.