# OpenReview forum: "Elicit and Enhance: Advancing Multimodal Reasoning in Medical Scenarios"
_ICLR.cc/2026/Conference — ICLR 2026 Conference Withdrawn Submission_

### Official Review · Reviewer_95j8 · 2025-10-29

**Soundness:** 3
**Presentation:** 3
**Contribution:** 2
**Rating:** 2
**Confidence:** 4

**Summary:**

This paper proposes $MedE^{2}$, a two-stage post-training framework to enhance multimodal reasoning skills of medical vision-language models. Stage I focuses on chain-of-thought-based supervised fine-tuning with curated medical reasoning process data, while Stage II applies multimodal preference alignment with DPO to improve logical consistency and reduce hallucination. Experiments on several medical benchmarks show consistent and notable performance improvements, especially for smaller models, without relying on reward models for medical reasoning. Overall, the topic is important and well motivated.

**Strengths:**

1. Clear motivation: The performance drop of LVLMs in medical reasoning tasks is convincingly demonstrated by empirical evidence.
2. Well-designed training framework: The two-stage strategy effectively avoids the difficulty of reward model construction in the medical domain.
3. High-quality data construction with professional validation significantly enhances training reliability.
4. Strong performance improvements, including outperforming much larger proprietary models, demonstrating good scalability.
5. Reduced hallucination and improved visual grounding, supported by both quantitative and qualitative results.
6. Good reproducibility: Code and model checkpoints are promised to be released.

**Weaknesses:**

1. Limited conceptual novelty: The method closely aligns with existing CoT SFT + DPO pipelines (e.g., DeepSeek-R1), with improvements mainly on the medical data side.
2. Unclear contribution of the visual modality: It remains uncertain whether the model truly leverages image features or merely relies on textual priors.
3. Small-scale medical data (5k) with unclear diversity and potential hidden curation bias.
4. Reliance on closed-source proprietary models for filtering and preference judging affects fairness and reproducibility.

**Questions:**

1. Can the authors quantify the causal contribution of visual inputs? (e.g., image-shuffling / masking experiments for visual dependency evaluation)
2. How robust is the preference alignment for rare disease reasoning?

---

### Official Review · Reviewer_ZoYc · 2025-10-30

**Soundness:** 2
**Presentation:** 2
**Contribution:** 1
**Rating:** 2
**Confidence:** 4

**Summary:**

This paper proposes MedE2, a two-stage post-training pipeline with DPO reinforcement learning designed to improve multimodal reasoning for medical large language models (MLLMs). In addition, the paper introduces an LLM-based multimodal medical reasoning dataset. The authors curate a 5K-sample dataset and evaluate MedE2 across several benchmarks, demonstrating strong performance.

**Strengths:**

1. The paper presents a clear and well-motivated rationale for developing multimodal clinical LLMs.

2. It introduces a carefully curated multimodal medical dataset.

3. The method leverages reinforcement learning through Direct Preference Optimization (DPO) to mitigate hallucinations in reasoning.

4. The proposed approach achieves superior quantitative performance compared to existing multimodal LLMs.

**Weaknesses:**

1. Limited contributions and conceptual overlap. The paper shows substantial conceptual overlap with prior work — ClinRagen [1]. The two-stage reasoning distillation framework, involving textual reasoning elicitation followed by multimodal enhancement, closely resembles existing work on clinical reasoning generation (e.g., ClinRagen). The contribution thus appears incremental or derivative, and the ClinRagen paper is neither cited nor properly acknowledged their contributions.

2. Lack of novel modeling components. The proposed method primarily combines supervised fine-tuning (SFT) and Direct Preference Optimization (DPO) within a standard pipeline, without introducing additional architectural or methodological innovations.

3. Quality concerns in dataset curation. The curated dataset may contain hallucinated reasoning samples generated by LLMs without expert guidance or verification, raising concerns about factual reliability.

4. Lack of qualitative evaluation. The generated reasoning should be qualitatively assessed by medical experts or using LLM-as-a-judge evaluation to verify clinical soundness and interpretability.

5. Generic ethical statement. The paper provides only a generic ethical statement, without discussing critical issues such as data usage ethics, medical safety, fairness, or potential reasoning errors.


Reference:


[1] Shuai Niu et al. Knowledge-Augmented Multimodal Clinical Rationale Generation for Disease Diagnosis with Small Language Models. In Proceedings of the 63rd Annual Meeting of the Association for Computational Linguistics (Volume 1: Long Papers), pp. 11011–11024, Vienna, Austria. Association for Computational Linguistics, 2025.

**Questions:**

See my weakness.

**Details Of Ethics Concerns:**

1. Limited contributions and conceptual overlap. The paper shows substantial conceptual overlap with prior work — ClinRagen [1]. The two-stage reasoning distillation framework, involving textual reasoning elicitation followed by multimodal enhancement, closely resembles existing work on clinical reasoning generation (e.g., ClinRagen). The contribution thus appears incremental or derivative, and the ClinRagen paper is neither cited nor properly acknowledged their contributions.

2. Lack of ethical discussion. The paper provides no meaningful discussion of ethical considerations, including data usage, medical safety, fairness, or potential reasoning errors.


Reference:


[1] Shuai Niu et al. Knowledge-Augmented Multimodal Clinical Rationale Generation for Disease Diagnosis with Small Language Models. In Proceedings of the 63rd Annual Meeting of the Association for Computational Linguistics (Volume 1: Long Papers), pp. 11011–11024, Vienna, Austria. Association for Computational Linguistics, 2025.

---

### Official Review · Reviewer_2hAd · 2025-11-01

**Soundness:** 2
**Presentation:** 2
**Contribution:** 2
**Rating:** 4
**Confidence:** 4

**Summary:**

The paper proposes MedE2, a two-stage post-training pipeline that first elicits structured reasoning via small, high-quality text-only SFT, then enhances and aligns multimodal clinical reasoning using DPO with a four-criterion preference judge to reduce hallucinations and improve image-grounded logic; with ~5K rigorously filtered samples, it yields consistent gains on MedXpertQA-MM and MMMU benchmarks, scales with model size, and outperforms SFT/RL baselines, with text-only elicitation proving most effective.

**Strengths:**

1. Clear, effective training recipe with small, high-quality data: Stage-I text-only elicitation reliably boosts complex clinical reasoning (including multimodal benchmarks) and scales better with larger base models; Stage-II DPO with MMRP further reduces hallucinations and promotes image-grounded, reflective reasoning.

2. Strong empirical validation and practicality: Consistent gains over competitive baselines and larger open-source models, robustness to inference-time scaling, careful data curation to reflect real clinical cases, and initial human–model agreement checks for the evaluator, highlighting real-world applicability to safer, explainable medical AI.

**Weaknesses:**

### Necessity of Stage-I and missing baselines

1. Many medical LLMs (e.g., Med-PaLM, BioMedGPT, LLaVA-Med) already show basic chain-of-thought (CoT) capabilities under prompting. Why is a separate text-only reasoning SFT (Stage-I) still necessary? If Stage-I is skipped and you go straight to multimodal training or preference alignment, what concrete negative effects do you anticipate (e.g., unstable reasoning chains, higher cross-modal hallucination, reduced utilization of image evidence, degraded long-chain consistency)?

2. Table 1 does not compare against established medical models. Could you provide results under a consistent evaluation protocol versus representative medical (or strong general) models? Alternatively, can you share an ablation comparing “No Stage-I → direct Stage-II” to quantify Stage-I’s gains on stability and performance (e.g., long-chain accuracy, cross-modal hallucination rate, error self-check hit rate)?

### Standards for the four preferences and evaluator reliability

3. For the four preference criteria (logical consistency, image-grounded reasoning, no hallucination, explicit reflection/self-check), do you have operational, reproducible definitions and scoring guidelines? For instance:

    1）Logical consistency: Are there formal contradiction checks or key-evidence consistency rules? Any labeling templates or decision trees?

    2）Image Analysis Involvement: How do you quantify genuine use of image evidence? Through explicit mention of visual findings (location, named signs, quantitative metrics) and alignment with annotations?

    3）No hallucination: How do you define factual errors and run detection? Do you have gold labels or inter-annotator agreement (e.g., κ) to report?

    4）Reflection/self-check: What minimal components are required (error modes, evidence re-check, ablation of alternatives), and how is it scored?

4. You use Gemini-2.5-Pro for evaluation. Given potential hallucination or bias in the evaluator itself, did you adopt multi-judge fusion (e.g., additional evaluators or human spot checks), report agreement across evaluators, or test robustness of results under different evaluators? If relying on a single evaluator, how do you mitigate evaluator–model coupling risk?

### Attribution of performance gains—data quantity vs. method quality

5. Did you run scale-controlled ablations to disentangle the effect of “more data” from the effect of “Stage-I design + preference alignment”? For example:

    1）Fix total sample size and compare randomly sampled open medical QA vs. your curated reasoning data.

    2）Under the same Stage-II, vary Stage-I size (e.g., 1k/3k) and composition (with/without reflection/self-check), then report performance.

    3）Provide curves showing how key metrics (multimodal accuracy, hallucination rate, long-chain stability, image-evidence utilization) change with data size and data type.

6. If full ablations are not yet available, could you share preliminary observations or failure cases (e.g., simply scaling generic QA increases templated answers, under-uses image evidence, or lacks reflection)？

**Questions:**

Please refer to the Weaknesses.

---

### Official Review · Reviewer_pKQj · 2025-11-03

**Soundness:** 2
**Presentation:** 3
**Contribution:** 2
**Rating:** 4
**Confidence:** 3

**Summary:**

This paper introduces $\operatorname{MedE}^{2}$, a two-stage post-training pipeline for advancing multimodal reasoning in medical domains. The approach first leverages curated text-only reasoning chains to elicit reasoning abilities in large multimodal models, followed by refining reasoning quality using Direct Preference Optimization (DPO) on a curated set of multimodal medical cases. Extensive experiments across medical reasoning benchmarks demonstrate significant improvements over strong baselines and opensource and proprietary large models. The authors provide detailed methodology, datasets, and ablation studies, as well as thorough qualitative analyses of real-world medical cases.

**Strengths:**

The paper presents comprehensive and rigorous experiments across a diverse suite of medical benchmarks, including MedQA, Medbullets, MedXpertQA-MM, MMMU-Health, and MMMU-Pro-Health. The pipeline is validated on multiple open-source base models (QwenVL2.5 and InternVL3.0 across several parameter scales) and compared against both opensource and proprietary state-of-the-art models (Table 1, Table 2), highlighting its robustness and scalability.

**Weaknesses:**

1. While the DPO loss is clearly presented (Equation in Section 3.3), the justification for why the particular Multimodal Medical Reasoning Preference (MMRP) criteria yield good calibration/hallucination mitigation is primarily empirical rather than theoretically grounded. A deeper theoretical or empirical comparison between different preference criteria, the effect of each criteria, or how MMRP compares to alternative alignment objectives (e.g., from MedMMV or LoRA-MedSim) would reinforce the technical soundness.
[1] Liu, H. et al., MedMMV (2025)
[2] Fahmy, K. et al., LoRA-MedSim (2025)
2. In constructing the MMRP-based preference pairs, there is some ambiguity around the selection and balance of preference samples, rejected versus accepted reasoning, and the exact application of Gemini-2.5-Pro for “selection” versus “scoring.” For example, does negative sampling prioritize specific error types? How sensitive are results to the mix of positive/negative pairs per case—what are the selection heuristics?
3. The ablation experiments are largely limited to model size scaling and Stage-I vs. Stage-II. There is limited analysis quantifying the isolated benefit of each MMRP criteria, the dependence on preference sample quantity or data modality breakdown, or how hallucination/consistency rates vary across sub-specialties or question types.

**Questions:**

Can the authors quantitatively isolate the effectiveness of each of the MMRP criteria (logical consistency, image analysis, hallucination, reflection) in the DPO pipeline? For example, ablations where only subsets of criteria are used, with performance on clinically relevant reasoning metrics?
How does the size and composition of preference pairs in Stage-II affect final performance and hallucination rates? Please clarify the procedure for sampling negative samples, and provide statistics (e.g., ratio of positive to negative pairs, distribution across question types).
Can more detail be provided about metric definitions for “reasoning accuracy” versus “understanding accuracy” in Table 1? How are these annotated, and how consistently are they scored for proprietary model baselines?
Can the authors elaborate on error types (e.g., as shown in Figure 9) observed after Stage-II? Are there systematic failure modes that remain, and how might further alignment or data curation address them

---

### Note · Authors · 2025-11-14

I have read and agree with the venue's withdrawal policy on behalf of myself and my co-authors.